# Upcycling Glass Waste into Porous Microspheres for Wastewater Treatment Applications: Efficacy of Dye Removal

**DOI:** 10.3390/ma15175809

**Published:** 2022-08-23

**Authors:** Sabrin A. Samad, Abul Arafat, Edward Lester, Ifty Ahmed

**Affiliations:** 1Advanced Materials Research Group, Faculty of Engineering, University of Nottingham, Nottingham NG7 2RD, UK; 2Department of Nuclear Engineering, Faculty of Engineering, University of Dhaka, Dhaka 1000, Bangladesh

**Keywords:** glass waste, wastewater treatment, upcycling, sustainability, circular economy

## Abstract

Each year about 7.6 million tons of waste glasses are landfilled without recycling, reclaiming or upcycling. Herein we have developed a solvent free upcycling method for recycled glass waste (RG) by remanufacturing it into porous recycled glass microspheres (PRGMs) with a view to explore removal of organic pollutants such as organic dyes. PRGMs were prepared via flame spheroidisation process and characterised using Scanning Electron Microscopy (SEM), X-ray diffraction (XRD), Brunauer–Emmett–Teller (BET) and Mercury Intrusion Porosimetry (MIP) analysis. PRGMs exhibited 69% porosity with overall pore volume and pore area of 0.84 cm^3^/g and 8.6 cm^2^/g, respectively (from MIP) and a surface area of 8 m^2^/g. Acid red 88 (AR88) and Methylene blue (MB) were explored as a model source of pollutants. Results showed that removal of AR88 and MB by PRGMs was influenced by pH of the dye solution, PRGMs doses, and dye concentrations. From the batch process experiments, adsorption and coagulation processes were observed for AR88 dye whilst MB dye removal was attributed only to adsorption process. The maximum monolayer adsorption capacity (q_e_) recorded for AR88, and MB were 78 mg/g and 20 mg/g, respectively. XPS and FTIR studies further confirmed that the adsorption process was due to electrostatic interaction and hydrogen bond formation. Furthermore, dye removal capacity of the PRGMs was also investigated for column adsorption process experiments. Based on the Thomas model, the calculated adsorption capacities at flow rates of 2.2 mL/min and 0.5 mL/min were 250 mg/g and 231 mg/g, respectively which were much higher than the batch scale Langmuir monolayer adsorption capacity (q_e_) values. It is suggested that a synergistic effect of adsorption/coagulation followed by filtration processes was responsible for the higher adsorption capacities observed from the column adsorption studies. This study also demonstrated that PRGMs produced from recycled glass waste could directly be applied to the next cyclic experiment with similar dye removal capability. Thus, highlighting the circular economy scope of using waste inorganic materials for alternate applications such as pre-screening materials in wastewater treatment applications.

## 1. Introduction

A key goal of the circular economy approach is to initiate sustainable manufacturing processes through conserving energy and natural resources [1,2,3]. This is plausible by minimising resource consumption, recycling, reclaiming and reusing or upcycling materials [4,5]. In 2018, the Environmental Protection Agency (EPA) published that those glasses alone contribute to 12.3 million tons of municipal solid waste (MSW) generation of which 3.1 million tons were recycled, 1.6 million tons were combusted and the rest (approximately 7.6 million tons) were landfilled [6,7]. It is estimated that MSW glasses will increase 13% by 2050 [2]. Waste disposal and emission have become of huge concern for areas with insufficient landfill capacities and inefficient municipal waste management (MWM) systems. An efficient MWM system relies on a financially sustainable, technically feasible, socially, legally acceptable and environmentally friendly system that is often unadaptable in most cases.

Glass is a vitrified inorganic material that is typically recovered from MSW at recycling facilities and is known as recycled glass (RG). RG consists of broken glass bottles, industrial and household flat glasses and other containers of clear and/coloured glasses. Only clear broken glass, known as cullets, can be recycled for reuse to produce new clear glass containers [8]. However, sorting clear glass from coloured glasses are too costly and time consuming process. The other well-known recycling process involves reusing waste glasses via vitrification [9]. However, this is often associated with greenhouse gas emissions (an estimated 16.9 MJ of waste heat and 0.57 kg of CO_2_ being produced for every 1 kg of glass sheet) [10]. Other less expensive treatments, including disposal in landfills or immobilisation in cement matrices, may become more attractive [10]. However, its execution necessitates a significant circular managerial approach with proper environmentally protective measures, sustainable material procurement, transportation and a green and sustainable design [11].

Water contamination produced by micropollutants has increased in recent decades, posing a significant threat to the environment and human development. This has fuelled the need for cost effective and sustainable water treatment options to meet present water quality demands. Activated carbons are made from carbonaceous precursors such as peat, coal and coconut shells and have been heavily used as adsorbents [12,13,14]. However, their high regeneration costs and specific legacy issues when fossil fuel are used as a source material can limit their applicability [15,16]. A large variety of alternate low-cost and environmentally friendly adsorbents such as (i) agriculture wastes, (ii) naturally occurring materials, i.e., wood, peat, coal, lignite, (iii) by-products such as sludge, fly ash, bagasse flyash, red mud, etc., have been examined with the aim to replace activated carbons by means of a sustainable material prepared via circular economy approach [17]. Relatively little research has been done with inorganic waste such as glass and glass ceramics for water treatment applications. Few studies have been reported to investigate silica glass frit as an alternative to sand in slow sand filters and rapid sand filters. Elif *et al.* reported that filters containing crushed glass reduced effluent turbidity and particle counts similar to those obtained with the sand filter. They showed that crushed glass had significant promise as an alternative to silica sand in rapid filtration [18], although the disadvantages associated with this filtration technique include reduced efficacy at low temperature and high turbidity, ineffective removal of coloured materials and imbalance in biological equilibrium due to the presence of toxic contaminants in raw water and loss of pressure head during operation [19,20].

Studies have shown application of glass as a substrate for immobilising photocatalysts such as TiO_2_ for photocatalytic water treatment [21]. A study conducted by Fernhndez *et al.* compared glass substrate with quartz and stainless steel substates for the photocatalytic degradation of malic acid [21]. The study concluded that quartz and stainless steel retained a constant photocatalytic activity of titania without deactivating TiO_2_ during the reaction to glass substates [21]. A study later conducted by Yu *et al.* demonstrated that this low photocatalytic activity of TiO_2_ was attributed to the diffusion of sodium and calcium ions from soda lime glass into the nascent TiO_2_ films [22].

Although the aforementioned applications used glasses without structural modification or post processing, only a few studies have used synthetic glasses or post processed waste glasses for wastewater treatment. The main reasons seem to be that the manufacturing is based on synthetic approaches which require use of expensive and or toxic chemicals, long processing time and expensive manufacturing approaches. More recently, Singh et al. reported 90% degradation of methylene blue dye using transparent silicate glass–ceramics containing LiNbO_3_ piezoelectric crystals, occurred within 150 min of ultrasonication due to a polarization field that forces free holes (h+)/electrons (e−) to participate in catalytic redox reactions [23]. These crystallites were fabricated by melt-quenching, followed by careful selection of heat-treatment time for controlled crystallization. However, the piezocatalysis phenomenon has been only observed in piezoelectric materials such as BaTiO_3_, ZnO, MoS_2_, LiNbO_3_, ZnSnO_3,_ etc. [23]. Generally, waste glass is composed of Borosilicate, aluminosilicate, and soda-lime based glass which would require doping it with such piezoelectric materials for piezocatalysis applications [10].

Nishida *et al.* reported on the fabrication of porous glass ceramic by sintering waste soda lime silicate and charcoal for water purification [24]. The porous glass ceramic produced from their study was successful in removing water turbidity and the pores maintained suitable environments for microbial proliferation. However, the manufacturing process of this porous glass ceramic was time consuming, as the waste glasses were thoroughly pulverized in a ball mill for more than 1 day, which was then sintered at temperatures of 800 °C to 1100 °C for 1 h or more. Furthermore, this approach failed to address the removal of organic and metallic pollutants from water or understanding its separation performance in the column process. A study reported by Hussain *et al.* for conversion of waste glass and mollusk shells into a porous material for separation of direct blue 15 azo dye from water using hydrothermal process [25] showed no data for column studies. There appears to be little published data on using silica in the column system. However, these silicas were primarily made from two types of silica sources: tetraethyl orthosilicate (TEOS) and sodium silicate solid or solution [26]. These raw materials, particularly TEOS, are prohibitively costly for large-scale nanoporous silicas manufacture [27,28]. Sheng *et al.* investigated the adsorption of methylene blue by silica using a column system [27]. However, the silica was synthesised by a two-step hydrothermal preparation process involving laborious and long processing time (36 h) and used toxic cetyltrimethylammonium bromide.

The current investigation involves waste glasses as an alternative sustainable source for wastewater treatment as they are nontoxic, chemically inert, and can be recycled and reused. Besides upcycling waste glasses for alternate applications would also minimize the use of natural resources for glass making, therefore maximising the service life of waste glasses by promoting reuse, recycling, and recovery [29]. Here, we demonstrate a route for upcycling/reusing of RG particles for potential application in wastewater treatment as a pre-screen material, to remove pollutants. RG particles were processed into porous recycled glass microspheres (PRGMs) via flame spheroidisation (FS) technique, which were then examined for removal of anionic Acid red 88 (AR88) and cationic Methylene blue (MB) dyes from water. In general, it is crucial to understand the surface charges of PRGMs in order to comprehend the overall dye-adsorption mechanism. In this context, point of zero charge and zeta potential measurements might be taken into consideration. However, it was challenging to investigate the surface charge characteristics of PRGMs due to their size limitation (55–180 µm). As such, using two different charged dyes would allow better understanding of the surface behaviours of PRGMs at different pH. The mechanism for dye removal and the reusability of PRGMs were also comprehensively investigated. One important aspect of this study was to eliminate the use of toxic and hazardous chemicals and offer a simple processing route for manufacturing RG into porous microspheres via use of our FS process, thus targeting reduction, reusability, and up-cyclability of inorganic waste materials.

## 2. Materials and Methods

### 2.1. Preparation of Microspheres

Porous recycled glass microspheres (PRGMs) were manufactured using a modification to the existing flame spheroidisation method called the Granular method. The Recycled glass (RG) particles used for this study were collection of mixed coloured waste glasses in size from 45–63 µm. In this method, RG particles below 45 µm size were mixed thoroughly with porogen (CaCO_3_, Sigma Aldrich, UK size ≤ 5 µm) with 1:3 *w/w* glass particles to porogen ratio and ground using a ball mill (Retsch PM 100, 20 small balls) for 15 min at 500 rpm. The ground particle blend was mixed with 2% PVA at ratio 2:1 *w/v* to form granules. The granules of size ≤ 200 µm were collected by sieving which were then flame-spheroidised using the thermal spray gun (MK74, Mettalisation, UK) at oxygen-acetylene ratio of 3:3. The microspheres manufactured were collected after 2 min from cooling trays. The PRGMs were then acid washed to remove any potential porogen remnants using acetic acid (5 M) for 2 min followed by washing in deionised water (for 5 min) and then dried in an oven at 50 °C for 24 h. The acid washed porous microspheres were denoted as W-PRGMs.

### 2.2. Characterisation Methods for RG and PRGMs

The morphologies of the RG and PRGMs were examined using FEI Quanta 600 scanning electron microscopy (SEM). PRGMs were resin embedded and then polished to investigate the internal structure.

X-ray diffraction (XRD) analysis for RG and PRGMs were explored with Da Vinci using a LYNXEYE XE-T detector in 1D mode using Ni-filtered Cu Kα radiation (λ = 0.15406 nm, 40 kV and 40 mA). The step time was maintained 5 s with a step size of 0.02°. Crystalline phases were identified using the EVA software (DIFFRACplus suite, Bruker-AXS) and the International Centre for Diffraction Data (ICDD) database (2005).

The specific surface area was then calculated from a relative pressure (P/P_O_) range of 0.05–0.25 employing the Brunauer–Emmett–Teller (BET) method. T-plot model (Harkins Jura correction) was used to calculate micropore volume and area, and BJH model (Harkins Jura correction) was applied to determine meso and macropore volume [30]. Total pore volume was calculated by adding both t-plot and BJH volumes together.

### 2.3. Dye Removal Studies

Methylene blue (MB) and Acid red 88 (AR88) dyes were considered as a model micro pollutant in this study. AR88 is an anionic dye while MB is a cationic dye. Acid red 88 (AR88) from ACROS Organics (USA) and Methylene Blue (MB) obtained from Sigma Aldrich (Dorset, UK) were made fresh before each experiment in deionised water.

#### 2.3.1. Batch Adsorption Experiments

The batch experiments were carried out using 250 mL Erlenmeyer flasks with 1 g/L PRGMs dosing and 50 mL dye volume for scoping studies. For isothermal investigations, the solution was agitated on a shaker (Orbital shaker, Thermo Fisher Scientific, Loughborough, UK) at 180 rpm at room temperature (22 ± 2 °C) with various concentrations of dye solution (10–300 mg/L) until equilibrium was obtained. PRGMs were isolated after adsorption using a 10 µm mesh sieve as a filter. A UV–VIS spectrophotometer (Varian Cary 50 UV-Visible spectrophotometer Agilent, Santa Clara, CA, USA) was used to quantify the dye concentration in solution at 505 nm (λ_max_). The adsorption capacity was calculated using the equation below (see Equation (1))
(1)$$q_{e\;}=\frac{\left(c_{0}-c_{e}\right)}{W}*V$$
where *C*_0_ is the initial concentration of dye (mg/L), *C_e_* is the equilibrium concentration which represents the remaining dye in the flask (mg/L); *V* is the volume of the solution used (L) and *W* is the weight of the PRGMs (g).

#### 2.3.2. Coagulation Study

In the second set of experiments, coagulation effect of PRGMs were investigated via Jar coagulation test. The optimum dose was chosen to be 1 g/L at pH 2–4 with dye concentration 200 mg/L for AR88 and 50 mg/L for MB dye. These parameters were chosen based on scoping studies. Experiments were conducted with 1 L glass jar using a magnetic stirrer (IKA, C-MAG HS 7, Loughborough, UK). The experimental procedure consisted of two subsequent stages: a rapid mixing stage at 200 rpm for 1 min, followed by slow mixing at 40 rpm for 20 min. The stirring was then stopped, and the mixture was allowed to settle for 6 h. Finally, the supernatant was removed from the beaker for UV analysis.

#### 2.3.3. Column Adsorption Experiment

The columns operational functionality and adsorption capacity of PRGMs was investigated using a column (Swagelok ¼″ inch) of 3.86 mm internal diameter by 200 mm height. A known quantity of microspheres was packed in the column to set the desired bed height. Dye solution of concentration 300 mg/L was fed using a pump (Gilson 305 Dunstable, UK) into the top of the column. Operational flow rate for the experiment was based on the empty bed residence time (EBCT) that is the total time needed to occupy the empty column, a parameter that influences adsorption capacity and breakthrough in the adsorption column [31]. Total adsorption was calculated by integrating the C_0_-C_t_ vs. t curve according to equation
(2)$$q_{total}=\frac{Q}{1000}\int_{0}^{t}C_{ad\;\;}dt$$
where *q_total_* (mg) is the total adsorption, *Q* (mL/min) is the flow rate, *C_ad_* (mg/L) is the adsorbed concentration (C_0_-C_t_), *t* (min) is the adsorption time. Equilibrium uptake (q_eq/_mg/g) in the column was calculated using *q_tot_* from equation [32]
(3)$$q_{eq}=\frac{q_{total}}{m}$$
where *m* (g) is the number of microspheres used.

Three commonly utilised models in the literature called Thomas, Adams–Bohart and Yoon Nelson models were explored for the kinetic studies [33]. The linearized form of the models is shown below as Equations (4)–(6), respectively
(4)$$\mathrm{Thomas}\mathrm{model}:\;\ln\left(\frac{C_{0}}{C_{e}}-1\right)=\frac{k_{Th}q_{Th}m}{Q}-k_{Th}C_{0}t$$
(5)$$\mathrm{Adams}\mathrm{Bohart}\mathrm{model}:\;\ln\left(\frac{C_{t}}{C_{0}}\right)=k_{AB}C_{0}t-k_{AB}N_{0}\frac{z}{U_{0}}\;$$
(6)$$\mathrm{Yoon}\mathrm{Nelson}\mathrm{model}:\;\ln\left(\frac{C_{t}}{C_{0}-C_{t}}\right)=k_{YN}t-\tau \;k_{YN}\;$$
where *C_t_* (mg/L) is the dye concentration at time, *t* (min), *C*_0_ (mg/L) is the initial concentration, *q_Th_* (mg/g) is the adsorption at capacity at exhaustion, *m* (g) is the mass of the adsorbent, *Q* (L/min) is the volumetric flow rate through the column, and *k_Th_* (mL/min/mg) is the Thomas rate constant. *k_AB_* (L/mg/min) is the Adams–Bohart constant, *N*_0_ (mg/L) is the adsorbate concentration at exhaustion, *z* (cm) is the column height and *U*_0_ (cm/min) is the flow velocity through the column, obtained by dividing the volumetric flow rate by the cross-section area. *k_YN_* (min^−1^) is the Yoon–Nelson rate constant and *τ* (min) is the time taken for 50% exhaustion to occur.

#### 2.3.4. Statistical Analysis

R^2^ and chi-square tests were performed to determine the suitability of various kinetic and isotherm models in the present study [34]. The equation is as follows.
(7)$$X^{2}=\overset{N}{\underset{i}{\sum }}\frac{{\left(q_{exp}-q_{cal}\right)}^{2}}{q_{cal}}$$
where, *q_exp_* and *q_cal_* (mg/g) are experimental and calculated dye concentration at equilibrium, respectively and *q_cal_* (mg/g) is average value of *q_cal_.*

#### 2.3.5. Recyclability Study of PRGMs

Regeneration and recyclability of dye adsorbed porous microspheres is a key characteristic that enhances sustainability of the adsorption process. Adsorbent regeneration was investigated only for AR88 dye as the dye was not the focus of the experiment, rather than the regeneration method and the subsequent performance of the porous microspheres.

A solution of AR88 (180 mg/L) was stirred at 293 K with 5 g/L of porous microspheres until equilibrium was reached. A 1 mL aliquot was taken for analysis and the adsorption capacity was calculated. In the first study, the dye adsorbed microspheres were heated in a furnace at 400 °C for 1 h to burn off the dyes from the microspheres. Thermogravimetric analysis (TGA) and XRD analysis were performed after heating in furnace to confirm the dye was completely removed. These desorbed microspheres were reused in the next adsorption cycle. The process was followed for 5 cycles.

### 2.4. Dye Removal Mechanism Investigation via Structural Analysis

#### 2.4.1. Fourier Transform Infrared Spectroscopy (FTIR) Analysis

Infrared spectroscopy of the RG and PRGMs were performed using a Brüker Tensor 27 spectrometer (Brüker Optics, Ettlingen, Germany) which was operated in absorbance mode. Spectra were measured in the region of 480 to 4000 cm^−1^ utilising a Standard Pike ATR cell (Pike Technologies, Inc., Essex, UK). OPUS software version 5.5 was used for the analysis.

#### 2.4.2. X-ray Photoelectron Spectroscopy (XPS) Analysis

XPS was performed using a VG Scientific ESCA Lab Mk2 electron spectrometer with a monochromatic Al KaX-ray source (1486.6 eV) operated at 15 mA and 10 kV anode potential. For the XPS measurement, samples were adhered to a carbon tab attached to an aluminium stub. The shift of the binding energy due to surface charging effect was calibrated by referencing the measured binding energy of C 1s to 284.8 eV. For high-resolution spectra, 20 scans were taken using pass energy of 20 eV. CASA-XPS software was used for data processing and curve fitting of XPS spectra. These spectra were deconvoluted by using a best fit programme with Gaussian distributions.

## 3. Results

### 3.1. Characterisation of RG and PRGMs

SEM images of irregular shaped as received recycled glass (RG) particles (≤63 µm) are presented in Figure 1a. It can also be seen from Figure 1b,c that a mixture of large porous (125–180 μm) and smaller dense (55–80 μm) microspheres were produced after processing the RG using flame-spheroidisation. The external pores of large PRGMs (125–180 μm) exhibited a pore size distribution (PSD) of 0–8 μm (analysed using Image J, see Appendix A). Cross-sectional analysis as presented in Figure 1d,e revealed porous internal microstructure. The surface area of RG and PRGMs were determined using N_2_ sorption analysis. It was apparent that the surface area of PRGMs increased after post processing (see Table 1). For example, the BET specific surface area of for PRGMs and RG were 8 m^2^/g and 0.03 m^2^/g, respectively. N_2_ sorption studies also provided detailed insights for micropores (≤2 nm), mesopores (2–50 nm), and macropores (0–140 nm). In Table 1, A_micro_ and V_micro_ represents pore area and pore volume respectively for micropores using t-plot model (Harkins/Jura correction) and V_meso_ and V_total_ represents pore volume of mesopores and total volume of macropores, respectively, using t-plot + BJH model (Harkins/Jura correction) [30]. It was observed that PRGMs exhibited micro, meso and macro pore volume after processing (see Table 1). However, the SEM images clearly indicated macropores larger than 140 nm. As a result, it was necessary to carry out Mercury Intrusion Porosimetry (MIP) analysis for macropore volume and size distribution to determine an accurate total pore volume as well as porosity of the PRGMs. The MIP study demonstrated that PRGMs exhibited large and smaller pores of diameter ranges from 5–0.450 μm and 0.450–0.017 μm, respectively as presented in Appendix A. The corresponding porosity profiles observed were 69% and 33% for PRGMs and RG, respectively. Furthermore, the total intrusion volume (overall pore volume) and pore area of 0.84 cm^3^/g and a pore area of 8.6 cm^3^/g were observed for PRGMs after processing using MIP analysis (see in Appendix A).

An X-ray diffraction pattern of the starting RG materials (see Figure 1f) revealed a single broad halo shaped peak between 15° to 39° of the diffraction angles. The absence of any sharp crystalline peaks suggested the amorphous nature of the RG particles explored. The XRD profiles of PRGMs presented in Figure 1f revealed peaks corresponding to the crystal planes (104) and (200) of calcium carbonate (CaCO_3_, JCPDS #01– 072– 1652) and calcium oxide (CaO, JCPDS #01– 074– 1226), respectively.

### 3.2. Dye Adsorption Studies

#### 3.2.1. Effect of Initial pH

The initial pH of the solution can significantly influence adsorption of dyes; thus, the effects of solution pH on dye adsorption using the PRGMs were first studied. A range of pH conditions were investigated to explore the effects of alkaline and acidic conditions (across the pH range from 2–12) on adsorption performance of PRGMs manufactured utilising both an anionic dye acid red 88 (AR88) and a cationic dye methylene blue (MB).

For the AR88 dye, maximum dye removal (~100%) was observed at the initial pH ~2.5 for PRGMs. As seen from Figure 2, the dye removal percentage decreased by approximately 10% as the initial pH increased from 4 to 8 and increased by approximately 4% beyond pH 8. On the contrary, a slight increase (by 2%) in dye removal was observed above pH 10 for the MB dye.

#### 3.2.2. Effect of Adsorbent Dose

The PRGMs dose (mg/L) was varied from 1 g/L to 20 g/L in order to investigate the effect of PRGMs mass loadings in terms of dye removal %. The effect of porogen on dye removal was also explored by washing the PRGMs (denoted as W-PRGMs) and using porogen alone. The solution pH was adjusted to 2 and 10 for AR88 and MB, respectively.

It can be seen from Figure 3a that the dye removal % increased with increasing PRGMs and porogen (CaCO_3_) dose from 1 g/L to 10 g/L for AR88 dye. The AR88 dye removal % for 1 g/L of PRGMs and porogen alone was 54 ± 2.1% and 53 ± 2.8% respectively, whereas for PRGMs of 10 g/L dose showed the dye removal values of 98.5 ± 1.5% and 98 ± 2% respectively. For W-PRGMs, the dye removal % was nearly the same for 2–5 g/L dosing which then decreased by 10% with increasing the dose to 8 g/L. However, on further increase of the W-PRGMs dose from 8 to 10 mg/L, the dye removal percentage increased slightly by 5%.

For MB, the dye removal percentage increased from 1 g/L to 6 g/L and then reached a plateau with no significant change in dye removal for PRGMs and W-PRGMs (see Figure 3b). In addition, no noticeable dye removal was observed by varying porogen dose for MB dye adsorption. For both dyes (AR88 and MB), the dye removal percentage of PRGMs was higher compared to W-PRGMs.

#### 3.2.3. Adsorption Isotherms

Adsorption isotherms are used to assess adsorbent–adsorbate (dye) surface interactions, particularly to estimate distribution of dye molecules (maximum uptake capacity, *q_m_*) at the adsorption sites. This capacity is related to fitting experimental data with given adsorption isotherms such as Langmuir and Freundlich models. Adsorption isotherms such as the Langmuir and Freundlich Isotherm are the most commonly found isotherms [35,36] as
(8)$$\mathrm{Langmuir}\mathrm{model}:\;q_{e}=\frac{q_{m}K_{1}C_{e}}{1+K_{1}C_{e}}$$
(9)$$\mathrm{Freundlich}\mathrm{model}:\;q_{e}=K_{f}C_{e}^{\frac{1}{n}}\;$$
where *q_e_* (mg/g) is the equilibrium adsorption, *q_m_* is the maximum dye capacity (mg/g), *C_e_* (mg/L) is the equilibrium dye concentration, *K_1_* the Langmuir equilibrium constant (L/mg), *K_f_* the Freundlich constant (L/g), *n* the heterogeneity factor. All the model parameters were evaluated by non-linear regression [37].

The isotherms for adsorption of AR88 and MB were calculated from experiments with varying dye concentration from 100 mg/L to 300 mg/L and 10 mg/L to 50 mg/L for AR88 and MB respectively (see Figure 4a,b) with a PRGMs dose 5 g/L. These parameters were chosen to ensure that PRGMs would be saturated so the maximum capacity of PRGMs from the isotherm experiments could be appropriately determined. The Langmuir and Freundlich models for AR88 and MB are illustrated in Figure 4. The parameters obtained from the models are outlined in Table 2.

From Table 2, comparing both Langmuir and Freundlich models, data for adsorption of AR88 on PRGMs fitted well to Freundlich’s model obtaining a higher correlation coefficient (R^2^_Freundlich_ = 0.99) closer to unity and smaller values of χ^2^ compared with those using Langmuir’s model. The maximum monolayer adsorption capacity (*q_m_*) calculated for PRGMs and porogen were 78 mg/g and 227 mg/g respectively. The isotherm studies were also conducted for unprocessed RG particles; however, no noticeable adsorption was observed for AR88 dye. On the contrary, it was not possible to carry out isothermal study for W-PRGMs for adsorption of AR88 due to formation of a coagulated mass, which was within 30 min (see Appendix A).

In contrast, adsorption of MB data fitted well with the Langmuir model (R^2^_Langmuir_ = 0.98) as values of χ^2^ were smaller compared to Freundlich model. The maximum monolayer adsorption capacity (*q_m_*) estimated for PRGMs was found to be 20 mg/g (see Table 2). Interestingly, the unprocessed RG or porogen alone did not show any MB dye adsorption.

### 3.3. Coagulation Study

It is apperent that W-PRGMs showed evidance of forming an agglomarated/coagulated mass for AR88 which hindered conducting kinetic studies (Appendix A). To simulate coagulation/agglormeration of AR88 dye, PRGMs with different doses were investigated using the commonly used the “jar test” method. It was observed that for different PRGMs dosing, the dye removal percentage was higher for W-PRGMs compared to PRGMs. For example, the maximum dye removal was observed as 91% ± 1.2 and 96% ± 1.6 for PRGMs and W-PRGMs, respectively (see Figure 5). With increasing PRGMs doses, the dye removal percentage values had not significantly changed for PRGMs and W-PRGMs. On the contrary, MB dye did not show any evidence of coagulation for both PRGMs and W-PRGMs.

### 3.4. Column Adsorption Studies

The column adsorption study is a more desirable and industrially practical for contaminants removal from wastewater [38,39]. The performance of fixed-bed column is studied using breakthrough curves. Figure 6 shows the concentration-time profiles obtained for fixed-bed column experiments at flowrates of 2.2 mL/L and 0.5 mL/L. The total amount of dye adsorbed and equilibrium adsorption of AR88 are outlined in Table 3.

From the data, an increase in adsorption compared to the batch process (adsorption capacity obtained from Langmuir isotherm) for PRGMs was clearly seen. Furthermore, the adsorption of PRGMs and W-PRGMs were higher than as received RG. The Thomas, Adams–Bohart and Yoon–Nelson models were applied to fit the experimental data to describe the fixed-bed column behaviour as shown in Appendix A. Between the Thomas and Adams–Bohart models, Adams–Bohart model was the least appropriate equation for modelling the adsorption in column experiment (see Table 4). The Thomas model provided the best fit for the experimental data at both flowrates (R^2^ > 0.80), suggesting its suitability for the design and scale-up purpose. Although the calculated adsorption capacity (*q_th_*) of microspheres was slightly lower than the experimental values (*Q_e exp_*).

Based on the Thomas model, the calculated adsorption (*q_th_*) capacities for PRGMs and W-PRGMs at flow rate 2.2 mL/min were found to be 250 mg/g and 244 mg/g, respectively and at flow rate 0.5 mL/min were 231 mg/g and 168 mg/g, respectively. The Thomas model rate constants increased with increasing flow rate; however, the equilibrium uptake capacity (*Q_e exp_*) decreased with decreasing flowrates. The Yoon-Nelson model decribes the half life of PRGMs before exhaustion i.e half life before regeneration [33]. It was observed that increased flowrate reduced the value of *τ_cal_*. However the *τ_cal_* value was lower than the experimental *τ_exp_* obtained from Yoon-Nelson model for both experimental flowrates.

### 3.5. Interaction of Dye with PRGMs

To examine the mechanism of dye interaction with PRGMs, analytical techniques such as FTIR and XPS studies were conducted. Figure 7 shows the FTIR spectrum of PRGMs before and after AR88 and MB dye adsorption in batch studies.

An intense adsorption at 1400–1500 cm^–1^, along with an adsorption peak at 874 cm^−1^ and 710 cm^−1^ were observed for porogen (CaCO_3_). These adsorption bands were assigned to stretching of C=O in CaCO_3_ and CO_3_^2−^ [40]. These peaks were also present in PRGMs before and after AR88 and MB dye adsorption. However, after dye adsorption a decrease in relative absorption intensity at ~874 cm^−1^ was observed for PRGMs+AR88 dye and PRGMs+MB dye (see Figure 7a,b) [40]. No noticeable peak shift or change in relative intensity was observed in the rest of the spectrum. The characteristic adsorption band for AR88 and MB dye were not detected in PRGMs+AR88 dye and PRGMs+MB dye. As a result, XPS studies were performed for further investigation, which investigated dye interaction on the surface of the PRGMs before and after dye adsorption. As shown in Figure 8, peaks corresponding to Si 2p, O 1s, C 1s and Ca 2p can be clearly identified in the survey scan spectrum for PRGMs before and after dye adsorption. No characteristic peak (N 1s, S 2s and S 2p) for AR88 and MB dye were identified on PRGMs after dye interaction. For more detailed analysis, high resolution scans were performed for PRGMs before and after dye adsorption for each signature characteristic peak (for C 1s, O 1s, Ca 2p and Si 2p).

Appendix A shows there were two carbon species present in C 1s spectra for PRGMs with the corresponding binding energies centred at 286 eV and 289.4 eV. The lowest binding energy represents the carbon on the carbon tab, whilst the highest binding energy is assigned to carbonate ion from the remnant porogen [41,42,43]. After dye adsorption, one new component of C 1s appeared at 290.8 eV which corresponds to π–π* transition in aromatic ring for the dyes [44,45]. Moreover, the peak at binding energy 286 eV and 289.4 eV showed deviation to higher binding energy after interaction of PRGMs with AR88 and MB dye.

O 1s spectra revealed characteristic peak for PRGMs at 535.8 eV and 533.3 eV for covalently bonded bridging oxygen (BO) atoms that connect two SiO_4_ tetrahedra; and non-bridging oxygen (NBO) atoms that covalently bonded to one SiO_4_ tetrahedron and ionically bonded to one alkali (or alkaline earth) ion surface oxygen and lattice oxygen, respectively [43,46] (see Appendix A). A shift to higher binding energy of O 1s spectrum was observed in PRGMs after AR88 and MB dye interaction. Furthermore, Appendix A showed that the relative peak intensity of NBO increased (with the BO component kept constant) after AR88 and MB dye interaction.

In Appendix A, Ca 2p spectra of PRGMs before adsorption was deconvoluted into four peaks: two peaks from Ca 2p_3/2_ and another two peaks from Ca 2p_1/2_. Ca 2p_3/2_ and Ca 2p_1/2_ with binding energy at 351.3 eV and 354.3 eV corresponds to Ca in CaCO_3_ and binding energy at 352.4 eV and 355.3 eV corresponds Ca in PRGMs. After AR88 and MB dye adsorption, the central peak for Ca 2p_3/2_ (351.3 eV) and Ca 2p_1/2_ (355.3 eV) also showed a peak shift to higher binding energy.

The Si 2p spectra in Appendix A represents two main features assigned to Si 2p_3/2_ at 102.7 eV corresponds to Si–O^−^ bond (silicon bonded with NBO) and Si 2p_1/2_ at 107 eV for to Si–O–Si bond (silicon bonded with BO) [46,47]. A closer look into Si 2p_3/2_ and Si 2p_1/2_ spectra revealed a shift of around 2.2–4.5 eV and 0.9–1.7 eV, respectively toward higher binding energy for PRGMs after dye adsorption.

### 3.6. Reusability of PRGMs

If the adsorbents developed could be regenerated and recycled after the adsorption process, then this would be hugely beneficial for potential sustainable material applications. The reusability of the PRGMs were investigated by repeating 5 cycles of the adsorption–desorption process in our study (see Figure 9). Adsorbent regeneration was investigated only for AR88 dye as the dye was not the main focus of the experiment, instead establishing a suitable solvent free regeneration process and investigating subsequent performance of PRGMs was the key outcome.

Initial PRGMs (control) showed AR88 dye removal percentage was 95 ± 0.8%. After the 1st cycle, the dye removal % decreased slightly to 87 ± 0.6%. For the rest of the cycles the dye removal percentage were fairly constant (86–87%). Similarly, PRGMs (control) presented MB dye removal percentage of 92 ± 1%. After the 1st cycle, the dye removal was 85 ± 0.6% as shown in Figure 9. These results suggested that heating the dye adsorbed PRGMs were an effective process for desorption. As such, DTG and XRD analyses were conducted further after heating in furnace to confirm the complete removal of the dye.

The DTG profile in Figure 10 revealed that AR88 and MB dyes were volatised below 400 °C. No traces of AR88 or MB were identified after dye desorption. The XRD patterns of desorbed PRGMs were found to be the same as the control PRGMs. However, a decreased peak intensity characteristic for CaCO_3_ and CaO were observed after desorption (see Figure 11a). Moreover, SEM images revealed that the pores of the PRGMs were closed after dye adsorption which then re-opened after the dye desorption process (see Figure 11b).

## 4. Discussion

The circular economy is a management or conceptual framework that prioritises material sustainability. Upcycling is sustainable in the sense that it helps to reduce waste in landfills, oceans, parks, and waste management facilities [10]. This is crucial as excessive waste can harm the environment if resource consumption is not properly managed based on the 6Rs (i.e., Reuse, Recycle, Redesign, Remanufacture, Reduce, Recover) [10]. The goal of upcycling is to create a product that is unique from the original yet has a higher economic value [48]. This paper demonstrated that reusing and/or upcycling of waste recycled glass (RG) for wastewater treatment applications was entirely feasible.

### 4.1. Processing of RG Particles into Porous Microspheres

RG mainly consists of a silica backbone with alkaline earth oxide glass former and/or modifiers. The SEM images revealed successful manufacture of PRGMs (see Figure 1b–e) via FS process. During flame-spheroidisation (FS), the prepared granules were fed into the high-temperature flame (∼3100 °C) where rapid melting and some melted glass droplet coalescence occurred. The molten particles spheroidised post exiting the flame due to surface tension. It is considered that porous structures formed due to decomposition of the porogen (CaCO_3_) in the form of CO_2_ gas bubbles while rapidly escaping at the point of solidification which led to the production of PRGMs exhibiting high levels of interconnected porosity. XRD results for the PRGMs indicated the presence of calcium carbonate (CaCO_3_) and calcium oxide (CaO), respectively. These porogen remnants, and some residual porogen most likely remained trapped inside the pores of the PRGMs post processing (see Figure 1f). In contrast, the RG remained amorphous.

The pore size determined using N_2_ adsorption desorption analysis showed that PRGMs formed micropores and mesopores as seen in Table 1. The BET surface area of the PRGMs was observed to increase compared to RG (from 0.03 m^2^/g for RG to 8 m^2^/g for the PRGMs). It is suggested that the increase in surface area post flame spheroidisation was due to porogen decomposition resulting in CO_2_ release, forming pores within the microspheres. For wastewater treatment studies, porous structure with decease in pore size is desired as most of the micro-pollutants are within size range of 10–0.001 µm [49,50,51]. Moreover, it was reported that pore size could also influence the rate of diffusion of pollutant molecules into the active site of an adsorbent [52]. As such, these pores will allow more separation of micro pollutants from wastewater [53].

### 4.2. Factors Influencing Dye Removal by PRGMs

PRGMs manufactured have shown that removal of Acid Red 88 (AR88) and Methylene Blue (MB) by PRGMs is influenced by dye solution pH, PRGMs doses, and dye concentrations.

The dye removal of PRGMs changed with changing pH for both dyes. Adsorption of AR88 by PRGMs showed an increased adsorption when the pH decreased (≤4) while adsorption of MB showed favourable adsorption at higher pH (≥10) as presented in Figure 2. The pH dependent adsorption performance of these dyes can be explained on the basis of pKa values of dyes. The pKa of AR88 is lower than 10.7 and pKa of MB is 3.8 which can be related with the possibility of existence of these dyes in anionic form at pH < pKa and cationic form at pH > pKa respectively [54,55]. AR88 is anionic dye that contains one sulfonic acid group (R-SO_3_Na). In aqueous solution, the AR88 dye dissociates to the sodium ion (Na^+^) and the sulfonate anion (R-SO_3_^−^) [56,57]. At acidic pH, the sulfonate groups of dye are negatively charged due to their pK_a_ values being lower than zero [56,57]. This negatively charged AR88 dye which might have electrostatically interacted with positively charged PRGMs (due to protonation) in acidic pH. When the pH of the dye solution increased, the number of positively charged sites on surface of PRGMs decreases. As in higher pH values, additional OH^−^ anions compete with the anionic AR88 for available active sites on PRGMs [58]. Therefore, there is a repulsive electrostatic force between the negatively charged surface of the PRGMs and negatively charged AR88 dye molecules. As a result, % AR88 dye removal decreases with increasing pH. Conversely, MB is a positively charged dye that could have electrostatically interacted with negatively charged PRGMs (due to de-protonation) in a basic pH.



**For the AR88 sorption in acidic medium**
PRGMs + H^+^ → PRGMs–H^+^            (Protonation of PRGMs)PRGMs–H^+^ + AR88^−^ → PRGMs–H−AR88

**For the MB sorption in basic medium**
PRGMs + OH^−^→ PRGMs–OH^−^          (de-protonation of PRGMs)PRGMs–OH^−^+ MB^+^ → PRGMs–OH−MB



The dosage of PRGMs was also an important parameter required optimisation to study the adsorption behaviour for removing dyes. The % dye removal efficacy of AR88 and MB dye increased from 1 g/L to 10 g/L with increasing PRGMs doses, and then reached a plateau with no significant change. The removal efficiency of dyes improved by increasing PRGMs loading due to the increased active sites within the PRGMs [58]. The dye removal reached a plateau when the available adsorption sites were greater than the adsorbate level (dye concentration) [59].

In isothermal studies, the R^2^ values of the Freundlich model was higher than Langmuir model for adsorption of AR88 dye. This indicated multilayer adsorption for the AR88 dye on the active sites within PRGMs [60,61]. Isothermal study indicated that porogen itself had an affinity to adsorb AR88 dye (see Table 2). PRGMs consists of approximately 55% of remnant porogen based on XRD (semi quantitative) analysis (see Figure 1f). This suggests that adsorption of AR88 could be due to porogen. However, it was seen that AR88 dyes were coagulated by W-PRGMs (PRGMs without remnant porogen) during isothermal studies. To simulate coagulation, Jar tests were conducted for W-PRGMs and PRGMs at same pH that was used for isothermal studies. It was observed after coagulation the % dye removal for W-PRGMs and PRGMs were ~96% and ~91% respectively as presented in Figure 5. Hence, PRGMs act as a coagulant and presence of porogen in PRGMs slightly decreased the coagulation process. Hence, the dye removal of AR88 dye by PRGMs were not solely attributed to porogen.

In contrast, the Langmuir model fitted well compared to the Freundlich model for adsorption of MB dye. This indicated homogeneous adsorption for the MB dye on the active sites within PRGMs [60,61]. Furthermore, isothermal studies showed that porogen has no affinity to adsorb MB dye. Thus, it could be suggested that adsorption of MB dye (see Table 2) was due to surface properties of PRGMs. The maximum monolayer adsorption capacity (*q_m_*) calculated for PRGMs was 78 mg/g and 20 mg/g for AR88 and MB dye, respectively. The adsorption capacity of AR88 was much higher than that of MB. The greater affinity of AR88 onto PRGMs than that of MB could be attributed to the chemical structure, surface charge as well as solubility of the dyes. In general, adsorption capacity on an adsorbent is influenced by molecular size and aromaticity, and by solubility, polarity and carbon chain branching in dyes [3]. Whilst both dyes are water soluble and AR88 has a larger molecular size than MB. Theoretically the smaller molecules would find more accessible pores for adsorption. However, AR88 is negatively charged dye with more benzene rings and accessible oxygen groups (see Figure 12), which might increase the adsorption capacity on PRGMs. In such regard, dyes of similar surface charge having different dye molecule size and surface groups for adsorption on PRGMs could be considered for future studies.

The column adsorption process requires prediction of the breakthrough curve for the experimental data through mathematical models for describing and analysing the lab-scale column studies for the purpose of industrial applications [64,65,66]. In this study, the breakthrough curves were fitted by linear regression to the Thomas, Adams-Bohart and Yoon-Nelson adsorption models for predicting dynamic behaviour of the column [67,68,69]. The Adams-Bohart model was the least appropriate equation for modelling the adsorption in these column experiments based on R^2^ values (see Table 4). This is primarily due to the fact that the Adams-Bohart model is best used for modelling early stages of adsorption and the lower end of the ideal breakthrough curve [67]. Therefore, the Adams-Bohart model cannot be considered useful for modelling the adsorption data in this research.

The Thomas model assumes a reversible pseudo 2nd order reaction and Langmuir adsorption characteristics and found a more appropriate fit to the experimental data [70]. Based on Thomas model it is possible to predict that adsorption would not be controlled by the chemical reaction, but by mass transfer at the interface [71,72].

Flow rate determines the sufficient contact between adsorbate and adsorbent. Hence considered as an important factor affecting the efficiency of the column [70,73]. The maximum adsorption capacity (*q_eq_*) of the microspheres using the column process increased with increasing flow rate as seen from the Thomas model. However, in most literatures the adsorption capacity of a column decreased with increasing flow rate [70,74,75]. Only a few authors reported increased column adsorption with increasing flowrate [76,77,78]. Column adsorption is based on complex phenomena such as axial dispersion, film diffusion resistance, intraparticle diffusion resistance (both pore and surface diffusion) and adsorption equilibrium with the adsorbent [79,80]. In a column process an adsorbate molecule must diffuse through the fluid film surrounding the adsorbent, travel by diffusion along the length of a pore until it finds a vacant active site for adsorption. In general, increasing the flow rate will decrease the film thickness and film resistance around adsorbent. Aroonwilas *et al.* reported increased adsorption contributes to the hydrodynamics of the column, which reflects the effective mass-transfer area (surface area) provided by packing [81]. Furthermore, Yu *et al.* while studying ammonia gas adsorption reported that the effective surface area of the column increases with increasing solvent flow rate [82]. It is expected that at a higher flow rate more turbulence within the column is created [83]. Under the turbulence mixing, the film around PRGMs would be thinner and more dyes could be transported to the surface of PRGMs [84]. Thus, there would be an increase in contact areas for mass transfer with increasing flowrates. Moreover, increased mass transfer/diffusion through inter-pore (inside microspheres) was achievable when the microspheres were more closely packed together [85]. Consequently, the overall adsorption capacity would increase with increasing flowrates, despite decreasing empty bed contact time (EBCT) or residence time (see Table 3). However, it is worth mentioning that the HPLC pump used for the experiment could not operate below 0.5 mL/min for an EBCT of 5 min. In this study, only two different flowrates were explored. To understand more, it would be highly recommended operating the adsorption column with varied flowrates to explore the influence of mixing rate in more detail.

The Yoon–Nelson model presented the 50% exhaustion time *τ,* which decreased with the increase in the flow rate as the saturation of the column occurred more rapidly. Moreover, from the Thomas model, the Thomas rate constant (*k_th_*) increased with increased flowrate indicating the column reached equilibrium more quickly. However, implications of the kinetics parameters identified from the Thomas model and Yoon–Nelson model are inconclusive due to the fact that none of the models fit well with the experimental data (<0.80). Significant discrepancies were observed between the calculated data and experimental data for column adsorption process (see Table 4). The Yoon-Nelson and Thomas models only provided an indication that increased flowrate and reduced EBCT could affect adsorption of AR88 onto PRGMs.

However, from the accumulated evidence, the proposed mechanism is column process is thought to be complex and could involve combination of several mechanisms including, adsorption and coagulation/filtration process [86]. Netpradit *et al.* reported while studying adsorption of reactive dye in a column system that in a down-flow system (where dye is flown in the direction of gravity), the adsorbent can serve for the adsorption and filtration of wastewater [87]. Qi *et al.* studied graphene oxide/chitosan sponge for column performance and stated that the material acted as a filtering material for removing MB from water while an adsorbent in batch process [88]. In another study reported by Dimitrova et al. mentioned coagulation effect of granular slag in column for lead removal [89]. PRGMs were explored as a coagulant for AR88. However, coagulation kinetics is much slower than the adsorption process. Thus, dye removal mechanism cannot result solely via coagulation. Furthermore, an increase in adsorption capacity was observed while comparing the batch adsorption process (Table 2 and Table 4), i.e., *q_e_* at batch study was 98 ± 4 mg/g, whereas in column testing *q_eq_* was <200 mg/g for PRGMs. In column process, the PRGMs are packed in the column providing more area of contact (interfacial area) for adsorption. Hence, more mass transfer/diffusion through interconnected pores (inside microspheres) would be achieved when the microspheres were more closely packed. Such packing will account for more accessible pores with an increase in effective surface area of contact for adsorption/filtration. Overall, it can be assumed that a synergistic effect of adsorption/coagulation followed by filtration process is responsible for showing discrepancy from the experimental value in column process.

### 4.3. Dye Removal Mechanism and Desorption Study

Besides active site interactions, adsorbent surface area, pore size distribution and pore structure of the adsorbent material also plays a decisive role in its adsorption properties [90,91]. In this study, to explore the interaction mechanisms of AR88 and MB dye with PRGMs, FTIR and XPS analyses of PRGMs post adsorption were conducted.

XPS high resolution scan for C1s of PRGMs after AR88 and MB dye adsorption revealed the presence of a new peak corresponding to C-O for AR88 and either C-N or C-S for MB. Moreover, the centre of C1s peak deviated to higher binding energy after adsorption. This would indicate possible dye interaction with the surface of PRGMs through C-OH linkers for AR88 and C-N/C-S terminal for MB dye (see Appendix A). Moreover, high resolution O1s spectra of PRGMs revealed a peak shift to higher binding energy with increased relative peak intensity of a non-bridging oxygen (NBO) after dye adsorption. Suzuki *et al.* reported hydroxyl group involved in the silicate network structure (Si–OH) can be classified as a NBO while studying low temperature porous glass fabrication by hydrothermal reaction [92]. As highlighted earlier, dye adsorption was influenced by change in pH. With an acidic pH, protonation (H^+^) of PRGMs might occur due to ionic exchange between the H^+^ of the solution and the modifying ions (Na^+^ and Ca^+2^) present in the glass network. It could be suggested that due to protonation of PRGMs the intensity of NBO increased according to reaction 10. As such, AR88 dye was adsorbed through hydrogen bonding via electro negative oxygen terminal of the dye and hydrogen on PRGMs.
–SiO^−^ +H^+^ **⇌** –SiOH(10)

At basic pH the surface of the PRGMs deprotonate which could result in possible hydrogen bonding via MB dye and the surface of PRGMs is enhanced based on the following reaction:–SiOH +OH^−^ **⇌** –SiO^−^ + H_2_O(11)

In high resolution Ca 2p spectra of PRGMs after AR88 and MB dye adsorption shown peak shift to higher binding energy for the central peak for Ca 2p^3/2^ and Ca 2p^1/2^. As illustrated in FTIR analysis, after dye adsorption a major change in relative absorption intensity at ~874 cm^−1^ was observed for PRGMs + AR88 dye and PRGMs + MB dye (see Figure 7). Both XPS and FTIR results indicated possible interactions of dyes with the porogen or Ca within silicate glass. However, it was evident that porogen does not have any affinity to adsorb MB. O1s spectra showed possible hydrogen bonding via Si-NBO site with MB dye. Thus, it is suggested that deviation to higher binding energy for Ca 2p was due to hydrogen bond formation with MB.

Furthermore, the Si 2p spectra revealed that there is a shift of around 0.9–1.7 eV and 2.2–4.5 eV, Si 2p_1/2_ and Si 2p_3/2_, respectively. Furthermore, the deviation for Si 2p_1/2_ (silicon bonded with BO) was higher than Si 2p_3/2_ (silicon bonded with NBO). This indicates possible electrostatic dye interaction with the surface of PRGMs.

Overall, the selective adsorption of AR88 and MB dye on PRGMs depends on its surface charge for electrostatic attraction. The mechanism of adsorption of dyes on PRGMs surface can be summarised in the following steps: in the first step, the H^+^/OH^-^ ions (pH dependent) interact with PRGMs and forms charged monolayer. Therefore, electrostatic attraction between oppositely charged surfaces was primarily responsible for adsorption of dyes. Besides electrostatic interaction, XPS studies also revealed possible Hydrogen bonding with the dye molecules (see Appendix A). Furthermore, the interaction of porogen with AR88 could be suggested during adsorption. However, for MB dye adsorption could have been a combination of many factors, such as surface area, surface hydroxyl density for possible hydrogen bonding and/or surface charge for electrostatic interactions [93].

Reuse of PRGMs is necessary to reduce costs as well as to preserve the environment. For this purpose, to recycle PRGMs, desorption of the dye and the reuse of the spent PRGMs (after desorption) in consecutive cycles were studied (see Figure 9). It was apparent that desorption via heating was proved effective for desorbing AR88 and MB dye. For both dyes PRGMs were heated in the furnace for 1 h at 400 °C. DTG analysis revealed that MB and AR88 dye completely decomposed after 350 °C (see Figure 10). The adsorption efficiency decreased after the 2nd cycle (% dye removal reduced to 87% from 95%) and remained constant for the reminder of the cycles (experiment was conducted for 5 cycles). This decrease in dye removal efficiency could be related to the decrease in adsorbent sites, the blockage of pores and loss of adsorbent after each adsorption recovery cycle [94]. However, SEM images revealed the opening of blocked pores after desorption. As such, one possibility would be reduced content of porogen after 1st adsorption cycle, as the desorption studies conducted for PRGMs contained remnants of porogen. Hence, after the 1st cycle, the adsorption process could have lost some remnant porogen. Moreover, PRGMs exhibited a melting temperature of ~1250 °C. Therefore, there is no possibility of losing PRGMs after heating. XRD results showed that after desorption no change of phases were visible (see Figure 11a). The reusability study conducted for MB suggested that a slight decrease in efficacy (% dye removal reduced to 85% from 92%). Although, it was evident from the isothermal studies that removal of MB was unaffected by porogen. Hence the decrease in efficacy could have been due to a result of blockage of active sites. However, it was inconclusive whether the dye removal performance would further decrease with each cycle or remain constant similarly to that observed for the AR88 dye based on reusability study conducted for 1st cycle. Therefore, to provide more insight into this investigation further studies are recommended. In general, heating adsorbed PRGMs indicated that PRGMs could be successfully reused several times.

Based on the overall findings, it was apparent that the PRGMs demonstrated promising dye removal efficacy in both batch and column process highlighting their capability to act as a coagulant, adsorbent or as filtration media. Here the PRGMs could be either used alone, as a coagulant/adsorbent before secondary treatment (biological treatment), or in conjunction with tertiary advanced treatments (such as adsorption processes) as a pre-screening material. However, in pilot-scale application for real wastewater the impact of such combinations would need to be thoroughly studied further.

## 5. Conclusions

This research was undertaken to explore manufacture of porous recycle glass microspheres (PRGMs) from recycled glass (RG) using flame spheroidisation technique and to evaluate its applicability for water treatment applications. PRGMs exhibited 69% porosity with overall pore volume and pore area of 0.84 cm^3^/g and 8.6 cm^2^/g, respectively (from MIP) and a surface area of 8 m^2^/g. The dye removal capability of PRGMs from water were investigated for two types of dye (AR88 and MB). A range of factors such as pH, adsorbent doses, dye concentrations influenced the adsorption kinetics and efficacy of the PRGMs. The optimum PRGMs doses used were 10 g/L and 6 g/L and pH were 2.5 and 7 for AR88 and MB dye, respectively. The maximum monolayer adsorption capacity (q_m_) calculated for UW-PRGMs were 78 mg/g and 20 mg/g for AR88 and MB dye, respectively.

The column adsorption study was conducted for RG, W-PRGMs and PRGMs for removal of AR88 dye. The study showed that the adsorption capacity increased with increasing flowrate which were 381 (±3) mg/g, 305 (±3) mg/g and 42(±3) mg/g for PRGMs, W-PRGMs and RG respectively. The breakthrough Thomas-Bohart and Yoon-Nelson adsorption models indicated that increased flowrate and reduced EBCT could affect adsorption of AR88 onto PRGMs. However, the experimental data have shown significant discrepancy with the calculated data. Thus, a synergistic effect of adsorption/coagulation followed by filtration process had been considered responsible for column adsorption process.

The reusability of the PRGMs were investigated by repeating the 5th cycles of the adsorption–desorption process. The PRGMs loaded with dye could be regenerated efficiently using 1 h of heating in the furnace at 400 °C. After desorption, DTG analysis confirmed that MB and AR88 dye were completely decomposed after 350 °C. PRGMs showed good recovery with almost stable efficiency (86–87%) for the second to fifth adsorption/desorption cycles. 

Overall, the current study addressed two environmental issues simultaneously, i.e., waste RG recovery, reuse, and recycling as well as how it may be utilized later to remove micro-pollutants for wastewater treatment, such as pre-screening materials.

## Figures and Tables

**Figure 1 materials-15-05809-f001:**
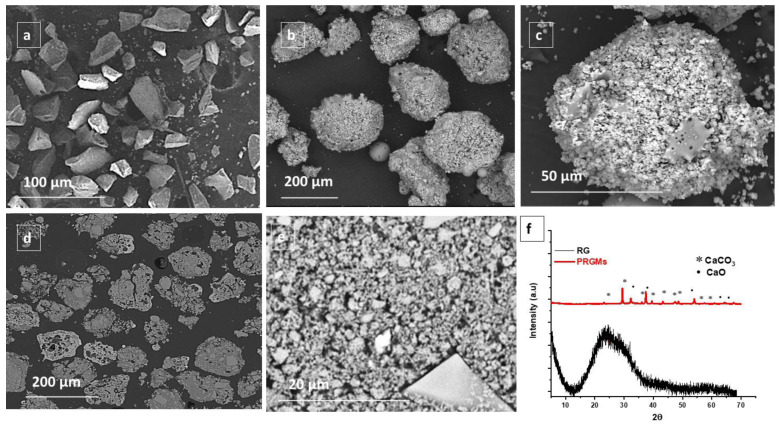
SEM images of (**a**) as received RG (particle size ≤ 63 µm), (**b**,**c**) surface morphology of porous recycled glass microspheres (PRGMs) at different magnification, (**d**,**e**) cross-section of PRGMs at different magnification and (**f**) XRD pattern of RG and PRGMs.

**Figure 2 materials-15-05809-f002:**
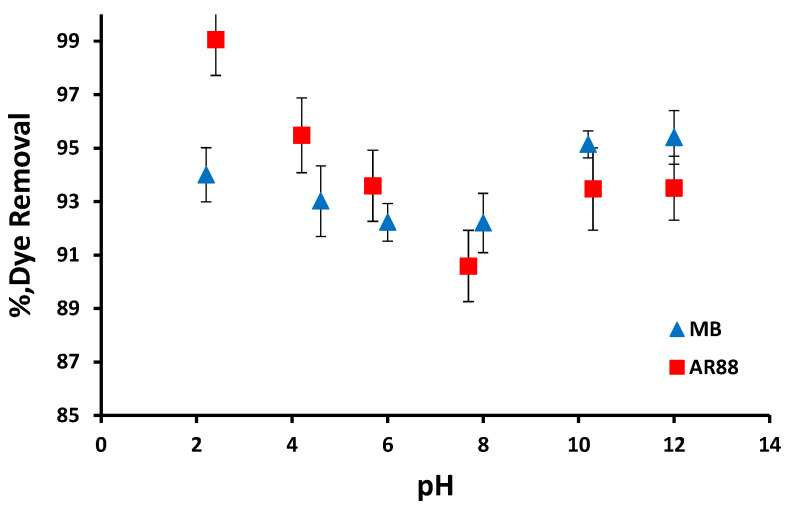
The effect of the initial pH of solution on the adsorption of AR88 and MB by PRGMs after equilibrium (C_0_ = 200 mg/L for AR88 and 25 mg/L for MB; dye volume = 10 mL; adsorbent loading = 10 g/L; T = 22 ± 2 °C).

**Figure 3 materials-15-05809-f003:**
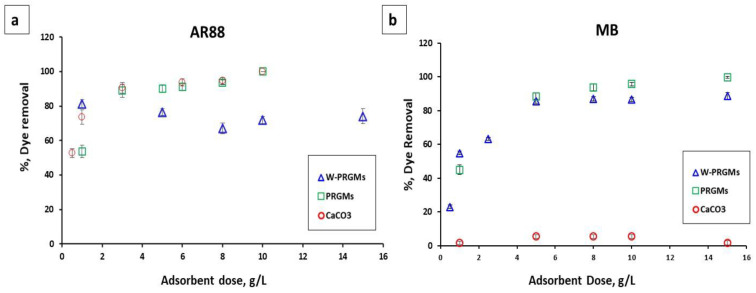
The influence of the PRGMs loading on the adsorption of (**a**) AR88 and (**b**) MB after equilibrium (C_0_ = 200 mg/L for AR88 and 25 mg/L for MB, pH = 2.5 for AR88, pH = 10 for MB; dye volume = 10 mL; T = 22 ± 2 °C).

**Figure 4 materials-15-05809-f004:**
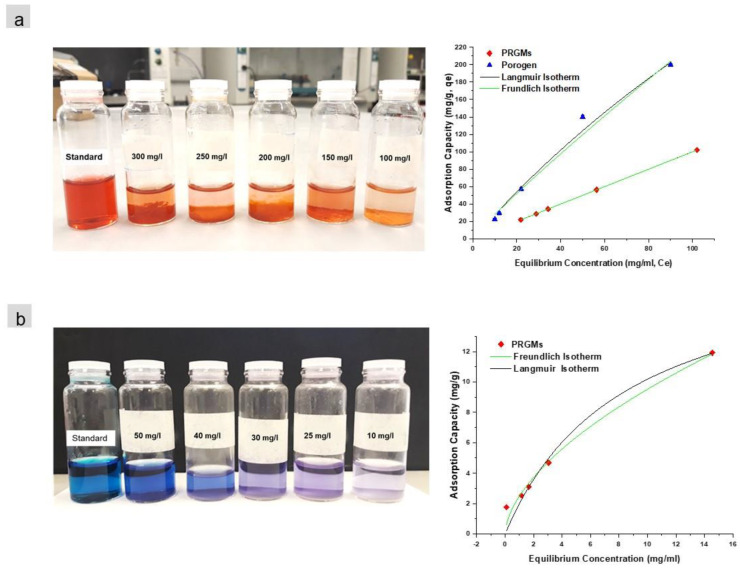
Isotherm data fitted with Langmuir and Freundlich models for (**a**) AR88 and (**b**) MB dye. (PRGMs dose = 5 g/L, pH 2.5 and pH 10 for AR88 and MB respectively and temperature of 22 ± 2 °C equilibrium time 24 h).

**Figure 5 materials-15-05809-f005:**
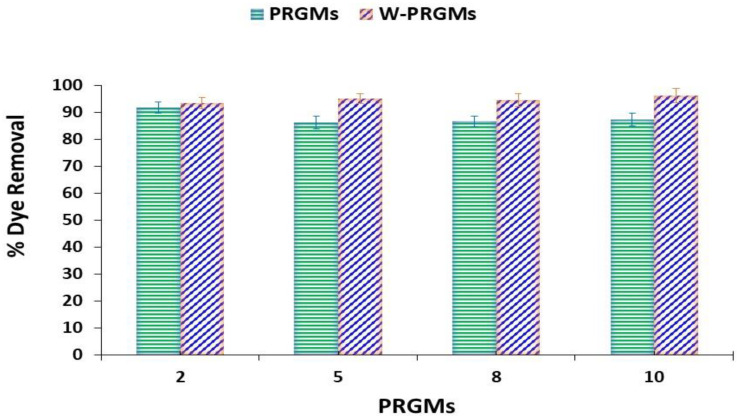
Effect of the PRGMs and W-PRGMs dose on removing AR88 dye by coagulation. (C_0_ = 200 mg/L, pH = 2.5, dye volume = 10 mL; T = 22 ± 2 °C).

**Figure 6 materials-15-05809-f006:**
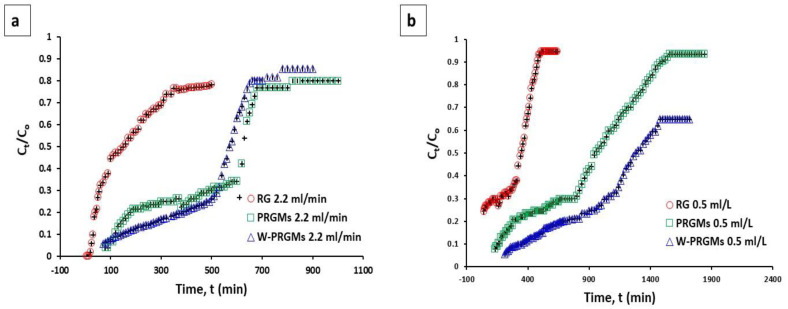
Breakthrough curve of AR88 dye adsorption for RG, PRGMs and W-PRGMs at (**a**) 2.2 mL/min and (**b**) 0.5 mL/min flow rates with starting dye concentration of 100 mg/L at 22 ± 2 °C.

**Figure 7 materials-15-05809-f007:**
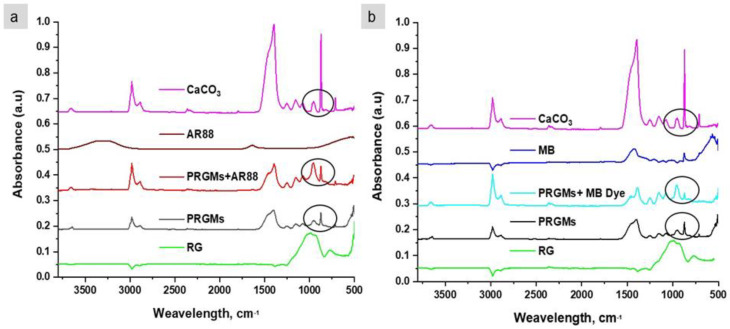
FTIR spectra for PRGMs before and after (**a**) AR88 and (**b**) MB dye adsorption.

**Figure 8 materials-15-05809-f008:**
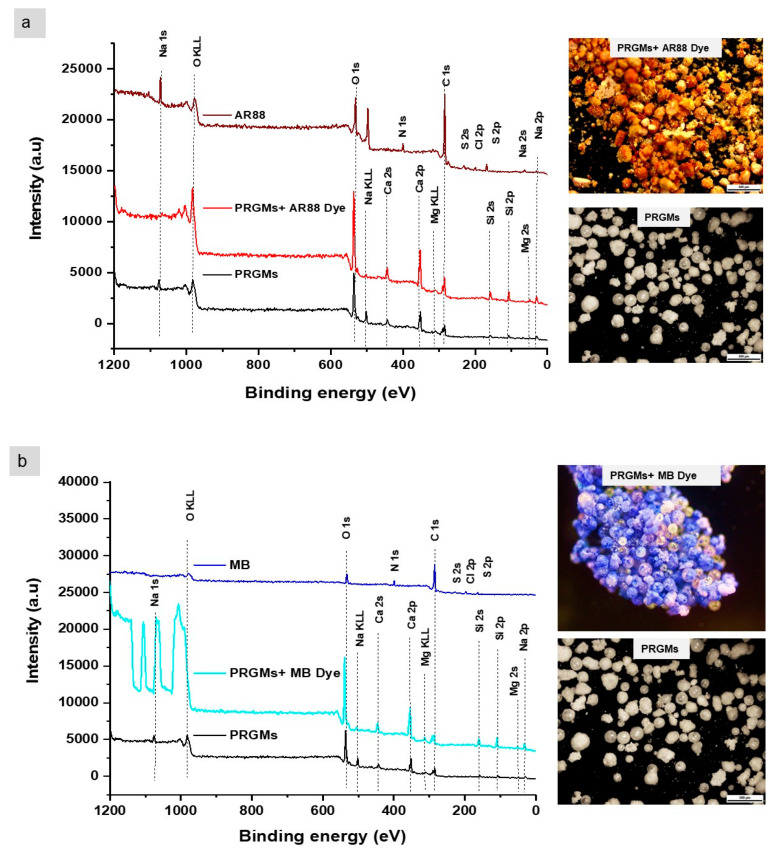
An XPS survey scan and optical microscopy images for PRGMs before and after (**a**) AR88 and (**b**) MB dye adsorption.

**Figure 9 materials-15-05809-f009:**
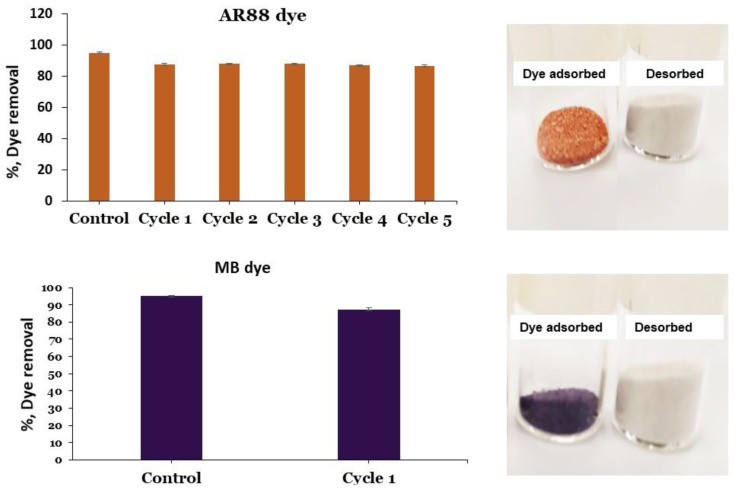
Recycling of the PRGMs for the removal of AR88 and MB (For AR88 C_0_ = 180 mg/L, Adsorbent dose = 5 g/L and pH = 2.5; MB, C_0_ = 25 mg/L, Adsorbent dose = 5g/L and pH = 7).

**Figure 10 materials-15-05809-f010:**
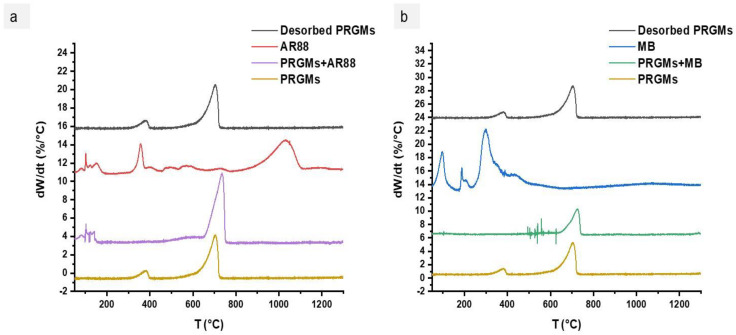
**A** DTG graph of PRGMs with (**a**) AR88 and (**b**) MB dye before and after dye adsorption and desorption.

**Figure 11 materials-15-05809-f011:**
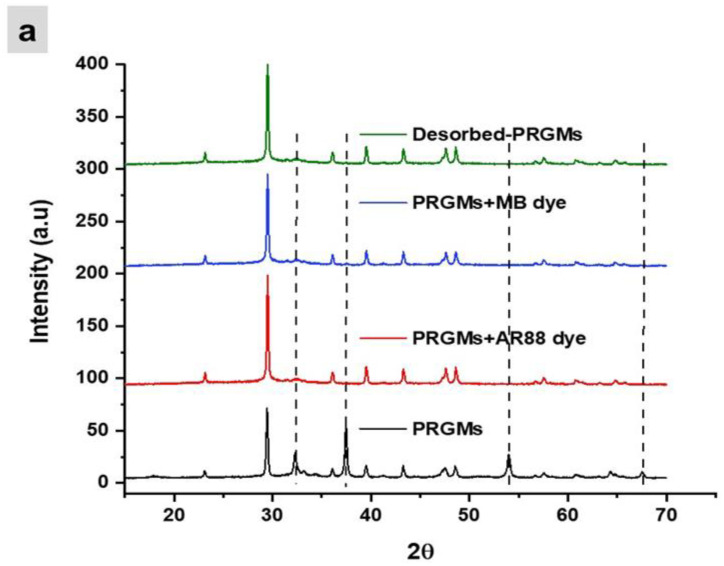
**The** XRD patterns (**a**) and SEM images (**b**) of PRGMs before dye adsorption, after dye adsorption and desorption.

**Figure 12 materials-15-05809-f012:**
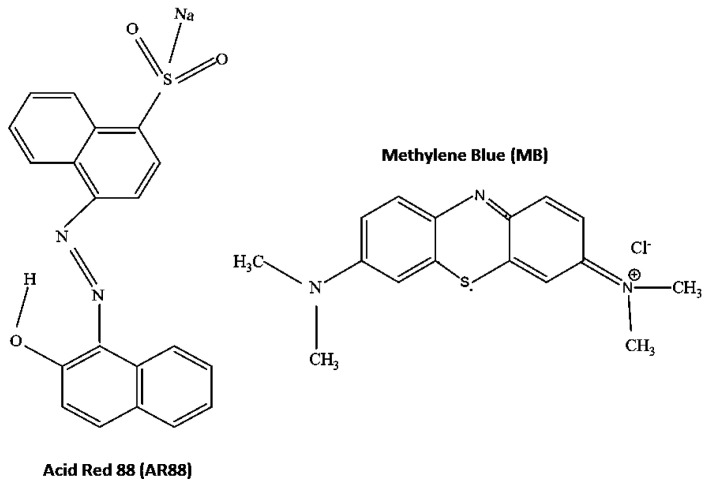
The structure of AR88 [62,63] and MB [63].

**Table 1 materials-15-05809-t001:** A pore size and pore volume analysis using t-plot and BJH method by nitrogen adsorption desorption analysis.

Sample	Surface Area (m^2^/g)	A_micro_ (m^2^/g)	V_micro_ (cm^3^/g)	V_meso_ (cm^3^/g)	V_total_ (cm^3^/g)
RG	0.03	0.006	0	0	0
PRGMs	8	1.5	0.002	0.03	0.06

**Table 2 materials-15-05809-t002:** Isotherm constants for adsorption of AR88 and MB.

		Langmuir Isotherm			Freundlich Isotherm				
**AR88 dye**	**Sample**	**q_m_ (mg/g)**	**R^2^**	**X^2^**	**q_e_ (mg/g)**	**K_f_ (mL/g)**	**1/n**	**R^2^**	**X^2^**
	**PRGMs**	78 ± 4	0.99	2.1	99	4 ± 0.01	0.71 ± 0.1	0.99	1.2
	**RG**	--	--	--	---	--	--	--	--
	**Porogen**	227 ± 5	0.99	3.07	200	2 ± 0.001	1 ± 0.2	0.99	5.3
**MB dye**	**PRGMs**	20 ± 3	0.98	0.37	14	3 ± 0.003	0.58	0.98	1.2
	**RG**	--	--	--	--	--	--	--	--
	**Porogen**	--	--	--	--	--	--	--	--

**Table 3 materials-15-05809-t003:** The total adsorption and equilibrium adsorption capacity of AR88 at different flow rates for RG, PRGMs and W-PRGMs.

Column Study					Batch Study
Sample	Flow Rate, (mL\min)	EBCT, (min)	q_tot_ (mg) (±3)	q_eq_ (mg\g) (±3)	q_e_ (mg\g) (±4)
**RG**	2.2	1	235	84	--
**RG**	0.5	5	89	42	
**PRGMs**	2.2	1	763	381	98
**PRGMs**	0.5	5	425	202	
**W-PRGMs**	2.2	1	611	305	--
**W-PRGMs**	0.5	5	282	157	

**Table 4 materials-15-05809-t004:** The breakthrough curve models with their parameters.

			Thomas Model			Adams-Bohart Model			Yoon-Nelson Model		
Sample	Flow Rate, (mL/min)	Q_e exp_, mg/g	q_th,_ mg/g	K_th_ (mL/min/mg)	R^2^	N_0_ (mg/L)	K_AB_ (mL/min/mg)	R^2^	τ_cal_ (min)	τ_exp_ (min)	R^2^
**RG**	2.2	84	12	0.001	0.64	39	4 × 10^−5^	0.92	175	204	0.94
	0.5	42	33	0.001	0.94	25	9 × 10^−5^	0.4	245	255	0.9
**PRGMs**	2.2	381	250	6.8 × 10^−5^	0.88	193	5.4 × 10^−5^	0.77	420	622	0.88
	0.5	202	231	3.1 × 10^−5^	0.80	135	2.2 × 10^−5^	0.61	800	914	0.86
**W-PRGMs**	2.2	305	244	9.5 × 10^−5^	0.83	222	5.4 × 10^−5^	0.7	400	549	0.83
	0.5	157	168	4.5 × 10^−5^	0.86	95	2.5 × 10^−5^	0.61	750	1285	0.80

## Data Availability

Not applicable.

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
