# Peer review of "Upcycling Glass Waste into Porous Microspheres for Wastewater Treatment Applications: Efficacy of Dye Removal"

_materials, 2022, doi:10.3390/ma15175809_

Round 1

Reviewer 1 Report

The manuscript deals with reusing/upcycling of waste recycled glass for wastewater treatment applications. Methylene blue and Acidic red 88 dyes were selected as the model pollutants. 

The manuscript is well-written and organized. The scientific results were explained and their importance was discussed in detail in the  Discussion part. 

There are some minor revisions/recommendations that need to be taken into account:

1) Different font type and size was used in some parts of the manuscript such as page 2-lines 53-56 and page 3-lines 142-145. The authors should do proofreading to correct these font mistakes and minor typos.

2) Page 4-Equation 1: In the equation, the denominator is "M" which usually denotes molar concentration, however in the text, "W" is defined as the weight of the PRGMs. 

3) Page 6-lines 245-246: The authors used the term "internal pores with porosity" which is not very clear. This term should be explained.

4) Page 10-lines 345-346 (Table 2): The authors claimed that the adsorption of MB on PRGMS data fits better with the Langmuir model with an R2=0.99. However, in Table 2, the R2 for Langmuir fitting is listed as 0.97, and Freundlich fitting is even higher than that with an R2 of 0.98. This claim needs further explanation.

5) Page 16-Figure 9: Although the adsorption/desorption of AR88 dye (reusability pf PRGMs) was studied conducting 5 cycles, the MB dye was studied with only 1 cycle. It is not clear whether the dye removal performance decreases with each cycle or remains constant as was the case for AR88 dye. 

Reviewer 2 Report

The work by Samada et al. reports on the solvent free upcycling method for recycled glass waste by remanufacturing into porous recycled glass microspheres, and their application in the removal of nano sized particles such as dyes from water was studied. The work is interesting, but some items are not clear and the content should be expanded.  

the comments are:

 1. Why a spherical shape was selected for dye adsorption? Moreover, the spherical shape of the PRGMs seems nor perfect from Fig.1, will this affect the functionality?

2. The reason for the increased pore and surface area of the PRGMs compared with the RG should be given. Besides, many adsorbing materials possess relatively high specific surface area for a good adsorption ability, how about the effect of the low specific surface area on the performance?

3. The formation mechanism and tunability of the microstructure of the PRGMs should be discussed for a better illustration of the advantages related to the specific material and structure.

4. Some minor mistakes should be eliminated such as particle size ≤63 in page seven.

Reviewer 3 Report

The manuscript is well written, and the experimental design is well planned and executed, and I have just one concern:

 1-   The authors wrote “explore removal of nano sized particles such as dyes from water.”, which is totally wrong as the dyes are totally soluble in water and does not considered as particles. The authors may write “removal of organic pollutants such as dyes”.

2-   Please enrich the abstract with some information about the Porous Microspheres characterization.

3-   Please elaborate more the XRD results as it is the only evidence for the transformation of the RG to PRGMs by indicating the JCPDs reference and the mechanism of the transformation of RG to PRGMs.

4-   Also, the manuscript missing important characterization such as the zeta potential or the point of zero charge to explore the surface charges of the RG, PRGMs, and W-PRGMs.

5-   Please discuss the effect of pH on the removal of both dyes by the PRGMs based on the pKa values of both dyes as well as the point of zero charge of the PRGMs. What is mentioned is not enough at all and no scientific explanation was used.

6-   Please be consistent with the abbreviation; either use porogen or W-PRGMs.

7-   Please discuss the removal of the target dyes by RG, PRGMs, and W-PRGMs. in real wastewater samples.

After a clear thought, I highly recommend the publication of the manuscript after taking into consideration the above mentioned comments.

Reviewer 4 Report

Dear Authors,

Your work deals with an interesting and important issue. However, it cannot be accepted in its present form, therefore I suggest some changes.

Firstly, I noticed different fonts, which might suggest copying fragments from other papers.

I would like to know why such dyes have been proposed. One is an azo dye (AR) and another - a thiazine dye (MB). Please add a short note in the Introduction section.

Please add some information about different type of sorbents for dye removal application and why you decide to use glass.

Please check line 522-527

Please expand Conclusions section.

Best regards,

Round 2

Reviewer 4 Report

Dear Authors,

Thank you for taking into account my comments.

Best regards